# Body Composition Analysis in Perimenopausal Women Considering the Influence of Vitamin D, Menstruation, Sociodemographic Factors, and Stimulants Used

**DOI:** 10.3390/ijerph192315831

**Published:** 2022-11-28

**Authors:** Dominika Kostecka, Daria Schneider-Matyka, Alina Jurewicz, Magdalena Kamińska, Katarzyna Barczak, Elżbieta Grochans

**Affiliations:** 1First Department of Ophthalmology, Pomeranian Medical University in Szczecin, Al. Powstańców Wielkopolskich 72, 70-111 Szczecin, Poland; 2Department of Nursing, Pomeranian Medical University in Szczecin, Żołnierska 48, 71-210 Szczecin, Poland; 3Department of Clinical Nursing, Pomeranian Medical University in Szczecin, Żołnierska 48, 71-210 Szczecin, Poland; 4Subdepartment of Long-Term Care and Palliative Medicine, Department of Social Medicine, Pomeranian Medical University in Szczecin, Żołnierska 48, 71-210 Szczecin, Poland; 5Department of Conservative Dentistry and Endodontics, Pomeranian Medical University in Szczecin, Powstańców Wlkp. 72 Av., 70-111 Szczecin, Poland

**Keywords:** body composition, perimenopause, vitamin D

## Abstract

(1) The perimenopausal period and menopause are physiological stages of a woman’s life, and they may result in the occurrence of many health problems. The aim of this study was to assess the impact of sociodemographic factors related to the use of stimulants and the presence of menstruation and vitamin D concentration in women’s health based on the analysis of their body composition parameters. (2) The study was carried out among 191 women. The diagnostic poll method was used, the levels of serum vitamin D were tested, and a body composition analysis was carried out. (3) Correlations between the vitamin D serum concentration and the following factors were established: adipose tissue mass (%) (rho = −0.18; *p* = 0.011), visceral adipose tissue (rho = −0.18; *p* = 0.014), BMI (rho = −0.22; *p* = 0.002), muscle mass (rho = −0.19; *p* = 0.008), osseous tissue mass (rho = −0.18; *p* = 0.013), and the phase angle value (rho = −0.2; *p* = 0.005). A statistically significant correlation between adipose tissue mass (Mdn = 34.4 vs. 32.2; *p* = 0.018; η^2^ = 0.029), visceral adipose tissue (Mdn = 8 vs. 6; *p* = 0.000; η^2^ = 0.106), and metabolic age (Mdn = 49 vs. 42; *p* = 0.000; η^2^ = 0.098) exists. (4) The following conclusions were made: (i) Menstruating women were characterized by increased body composition parameters, especially adipose tissue mass, visceral adipose tissue, and metabolic age. Greater muscle and osseous masses were noted in regularly menstruating women. (ii) A correlation between the vitamin D concentration and body composition parameters in the studied women was observed.

## 1. Introduction

The perimenopausal period and menopause are physiological stages of every woman’s life. Menopause is defined as the last menstrual bleeding which did not reoccur in the next 12 months and is not caused by any pathological conditions [1]. As indicated by the WHO, menopause occurs at around 50 years of age and is associated with a number of changes in the physical and mental condition of a woman. A slow termination of the ovarian hormonal function takes place, which causes the reorganization of the hormonal system (especially estrogen deficiency) as well as changes in the woman’s body composition. The changes are mostly related to the increase in the visceral adipose tissue mass. The perimenopausal period and menopause may result in the occurrence of many health problems and influence self-esteem [2]. Other co-occurring symptoms in the perimenopausal period include sexual dysfunctions, connective tissue disorders, and soft tissue and skin disorders. The premature incidence of menopause symptoms may be associated with genetic predispositions, menstruation disorders in the reproductive period, operations conducted on the reproductive organs, environmental factors (air pollution), the use of stimulants (smoking tobacco, drugs), and exposure to toxic factors (anorexia, bulimia, malnutrition). Delayed menopause may be caused by excessive body mass, obesity, alcohol consumption, or multiple pregnancies and births; however, the occurrence of menopause is not influenced by race, height, heavy physical work, or by the age at which the first period is experienced [3,4,5,6,7,8].

Nowadays, the average life expectancy for women ranges between 76 and 80 years. This higher threshold results from the development and achievements of modern medicine as well as improvements in everyday functioning. It is worth noting that the successive elongation of the average life expectancy among women has also caused prolongation of the postmenopausal period, which now constitutes at least a third of a woman’s lifespan. This extremely important fact deserves special attention from the whole healthcare system and calls for the introduction of appropriate diagnostics, prophylaxis, and therapy among the female population embarking upon this difficult time known as the perimenopausal stage. The ailments that can trouble women approaching menopause affect every aspect of their lives—not only the physical aspects but also social and mental ones [9]. It is worth noting that the process of aging is a natural part of human life; however, it rapidly increases after menopause. Thus, one of the most important tasks in human development is to accept the biological changes that take place inevitably and naturally. The perimenopausal period is a developmental stage that occurs appropriately when a woman consents to the slow termination of menstruation, and it is associated with physiological changes in her body. Only the acceptance of the transition into middle age might give a woman balance in all aspects of her life [10].

Lifestyle as a behavioral factor might prevent or advance the prevalence of many chronic diseases. While aging, people are more prone to diseases; thus, it is vital to maintain good mental and physical health, especially during the perimenopausal period. A stable lifestyle, healthy eating habits, and a balanced diet can slow down the aging process [11]. A healthy and active lifestyle, meaning regular meal consumption, avoiding stimulants, and partaking in physical activity, may lead to an improved quality of life and help maintain good health and well-being [12].

Dietary deficiency and inadequate exposure or reactivity to sunlight (due to lifestyle choices, cultural customs, and/or aging) were identified as important risk factors for vitamin D deficiency. The prevalence of a low serum 25(OH)D concentration (<50 nmol/L) is high, which occurs in more than 50% of people during winter in many European and Middle Eastern countries [13]. Vitamin D deficiency (25(OH)D < 20 ng/mL) was reported in 60.2% of women aged between 30 and 65 years living in Riyadh, Saudi Arabia [14]. Among young women with different body weights, 77.92% had low 25(OH)D levels (LD < 30 ng/mL) [15]. In a study by Schmitt et al., the serum 25(OH)D levels were sufficient in 32% of women and insufficient or deficient in 68% of women. Additionally, women with low 25(OH)D levels had higher TC, triglycerides, insulin, and HOMA-IR levels (*p* < 0.05) [16]. In the study provided in European countries, it was stated that there was a highly significant difference of 25(OH)D levels across European countries (*p* < 0.0001) among women in the postmenopausal period. The lowest level of 25(OH)D was found in France and the highest in Spain. In the whole study population, the prevalence of 25(OH)D inadequacy was 79.6% and 32.1% when considering cut-offs of 80 and 50 nmol/L, respectively; when considering patients aged less than 65 years, the prevalence reached 86% (cut-off of 80 nmol/L) and 45% (cut-off of 50 nmol/L) [17]. The prevalence of vitamin D inadequacy in studies of postmenopausal women in Eastern Asia ranged from 0 to 92%, depending on the cut-off level of serum 25-hydroxycholecalciferol [25(OH)D] that was applied (range ≤ 6–35 ng/mL (≤15–87 nmol/L)). One large international study found that 71% of postmenopausal women with osteoporosis in Eastern Asia had vitamin D inadequacy, defined as serum levels of 25(OH)D < 30 ng/mL (75 nmol/L) [18].

The aim of this study was to assess the impact of sociodemographic factors related to using stimulants, the presence of menstruation, and vitamin D concentration on women’s health in the perimenopausal period based on the analysis of their body composition parameters.

## 2. Materials and Methods

The study was carried out among women living in north-western Poland. The respondents were recruited via information on posters displayed in primary care facilities and published on social platforms. The willingness to participate was reported by phone.

In total, 191 respondents were included in the study. The following inclusion criteria were adopted:Perimenopausal age (45–65 years);Gender: female;A lack of vitamin D supplementation;Expressing a willingness to participate in the study;Signing an informed consent form for participation in the project;A lack of current cancerous, psychiatric, or inflammatory diseases.

Those who failed to meet these criteria were excluded.

The study was completed within a month, in October, right after the summer had ended. The study consisted of three stages. In the first stage, the diagnostic poll method was carried out using a questionnaire form. The questionnaire form constituted the author’s own questionnaire, which regarded sociodemographic data, chosen medical information, and matters regarding using stimulants (cigarettes, alcohol). The surveyed individuals answered questions regarding stimulants—in the case of cigarette use, respondents had to indicate whether or not they smoked cigarettes. In the case of alcohol consumption, less than 20 g of pure alcohol per day or sporadic consumption of more than 40 g of pure alcohol and a declaration of at least two days of abstention per week were adopted as a norm [19].

In the second stage, a peripheral blood sample was drawn to assess the level of vitamin D in the serum. To interpret the test results, the range of biological reference values for vitamin D according to the “Diagnostyka” laboratory, which has the quality certificate (ISO 9001:2015), were used:Vitamin D 25-OH D metabolite:
○Deficiency: <10 ng/mL,○Results below the norm: 10–29.99 ng/mL,○Results within the norm: 30–80 ng/mL,○Results above the norm: 80.10–99.99 ng/mL,○Toxicity: >100 ng/mL.

Biologic material samples were collected by trained nurses. Blood was drawn in a surgery room adapted to this procedure, applying the existing rules. Respondents who were included in the study fasted the day of the test. Each patient had 4 mL of blood drawn from a peripheral vein into a Vacoutainer-type test tube closed system. The obtained material was transported to the hospital laboratory “Diagnostyka”, in compliance with the procedural requirements for transporting medical samples, where the analysis was carried out.

Subsequently, the third part of the study consisted of individual body composition analysis with a Tanita MC780 MA body composition analyzer as well as having the measurements of the hips and waist taken by a qualified nurse team. Body composition was assessed using bioelectrical impedance analysis (BIA), which consists of evaluating the resistance of the flow of an electric current. For the BIA analysis, knowledge is used concerning the prevalence of electrolytes and better electrical conductivity of muscle tissue, which contains a considerable amount of water; in turn, adipose tissue is less conductive. The BIA is a reliable, non-invasive, and easily available means for the estimation of body composition parameters [20]. The results of the measurements allowed the following variables to be obtained: body mass (kg), body mass index, fat mass (kg), fat mass (%), visceral fat, metabolic age, muscle mass (kg), osseous mass (kg), water content (kg), and water content (%) and phase angle (°). Participants were instructed to avoid vigorous exercise at least 24 h before the test, finish the last meal at least 2.5 h before the measurement, empty their bladder before the measurement, and remove all metallic objects (e.g., jewelry and keys). Patients stood on the device with both feet parallel to the electrodes without bending their knees.

Participants who took part in the study were informed about its aim and the possibility of withdrawing from the study at any stage. All the respondents received test results with their interpretation keys.

The study received approval from the Bioethics Commission of Pomeranian Medical University in Szczecin (KB−0012/119/17).

The results obtained from the questionnaire forms, individual components of body composition analysis, and results of laboratory tests were coded and transferred to a Microsoft Office Excel spreadsheet (2016 version).

The obtained data were analyzed with descriptive statistic methods. Depending on the type of measuring scale used to express variables, two types of descriptive statistics parameters were used:For quantitative variables, measurements of the central tendency were calculated (mean and median) as well as dispersions (standard deviation and coefficient of variation); moreover, the minimal and maximal values were calculated;For nonmetric variables (qualitative and ordinal), the structure measure was established—number and frequency [21].

Comparison of quantitative values for two independent variables was carried out with a Mann–Whitney non-parametric test. This test is used to compare two sample means that come from the same population; it is used to check whether or not two sample means are equal. The ANOVA Kruskal–Willis range test was carried out for comparison of quantitative values for more than two values. It is a non-parametric method for testing whether samples originate from the same distribution [22]. Correlations between quantitative variables were described with Spearman correlation co-efficient range (rho-Spearman), a non-parametric measure of the rank correlation (statistical dependence between the rankings of two variables) that assesses how well the relationship between two variables can be described using a monotonic function [23].

Analysis was conducted with the statistics package STATISTICA, version 13.3 (TIBCO Software Inc.). For all of the analysis, zero hypothesis verification was carried out with an a priori adapted level of statistical significance equaling 0.05.

## 3. Results

Women’s average age was 53.1 ± 5.37, with a median of 53 years. The age of the respondents ranged between 45 and 65 years.

More than half of the respondents (55%) received higher education, and 38.7% had secondary education; 77.5% lived in cities with more than 100,000 residents; the majority (71.2%) were in a formal relationship; 88% were professionally active; 58.6% of the respondents were diagnosed with at least one chronic disease; 37.7% menstruated, 68.1% of whom had regular cycles, and 62.3% were already not menstruating; 37.7% of women reported frequent alcohol use; and 16.2% admitted to smoking tobacco (Table 1).

The height of respondents ranged between 150 cm and 180 cm. The average height was M ± SD = 164.8 ± 5.76 cm. The coefficient of variables for the body weight of the surveyed women was statistically significant and indicated 19.2%, which means that the body mass significantly varied among respondents. Woman’s weight varied from 44.4 kg to 115.4 kg. The waist circumference of those surveyed ranged from 61 cm to 129 cm, while the hip circumference ranged from 75 cm to 136 cm. The average waist circumference was M ± SD = 88.7 ± 12.87 cm, while the average hip circumference was M ± SD = 102.5 ± 10.04 cm (Table 2).

The lowest vitamin D concentration in women was 7.2 ng/mL, and the highest was 49.7 ng/mL; the median was 23 ng/mL, and the mean was 23.4 ng/mL. The coefficient of variables amounted to 34.6%. The level of vitamin D was below the norm in 78% of respondents (Table 3).

The lowest metabolic age was 30 years, while the highest was 74 years. The water content had the lowest volatility in the studied women (CV = 9.1%). The highest volatility was observed in terms of the adipose tissue mass parameter (CV = 35.3%). The lowest content of muscle mass amounted to 34.1 kg, while the highest was 64.2 kg. The most similar results were obtained during osseous mass analysis in the studied women (SD = 0.3 kg). The least similar results were recorded in the metabolic age (SD = 11.41 years). The highest median was 47.1% and regarded the water content of the studied participants (Table 4).

The age of respondents was positively correlated with the visceral adipose tissue (rho = 0.38; *p* = 0.000) and metabolic age. This means that an increase in both parameters correlates with age. The age was vaguely negatively correlated with the water content (rho = −0.14; *p* = 0.046). No statistically significant correlation was established between age and adipose tissue mass (*p* = 0.06), BMI (*p* = 0.129), muscle mass (*p* = 0.762), osseous tissue mass (*p* = 0.829), and phase angle values (*p* = 0.218) (Table 5).

The data analysis showed statistically significant differences (*p* < 0.05) between different parameters of body content analysis (adipose tissue mass, visceral adipose tissue, metabolic age, BMI, muscle mass, osseous tissue mass, water content, and phase angle values) and the surveyed participants’ level of education. The greatest influence in terms of the level of education was observed in terms of visceral adipose tissue and BMI (η^2^ = 0.046) (Table 6).

No statistically significant differences were established in the body composition analysis in relation to the place of residence (Appendix A), marital status (Appendix A), and professional activity (Appendix A) in the surveyed women.

A statistically significant correlation between adipose tissue mass, visceral adipose tissue, and metabolic age, as well as the water content in the body and the presence of menstruation in the group, were established. Values of all parameters were higher in the group of non-menstruating women than in menstruating women: adipose tissue mass (Mdn = 34.4 vs. 32.2; *p* = 0.018; η^2^ = 0.029), visceral adipose tissue (Mdn = 8 vs. 6; *p* = 0.000; η^2^ = 0.106), and metabolic age (Mdn = 49 vs. 42; *p* = 0.000; η^2^ = 0.098). The only exception was the water content—the value of this parameter was higher in the group of menstruating women than in the non-menstruating women (Mdn = 48.4 vs. 46.4; *p* = 0.006; η^2^ = 0.04). For the other parameters, no statistically significant differences were established (Table 7).

A statistically significant relationship between the muscle mass and osseous tissue mass and the regular menstruating cycle was established. Values of both parameters were higher in the group of regularly menstruating women than in the group of women who did not menstruate regularly: muscle mass (Mdn = 45.8 vs. 43.2; *p* = 0.016; η^2^ = 0.079); osseous tissue mass (Mdn = 2.5 vs. 2.3; *p* = 0.031; η^2^ = 0.08). For all the other parameters, no significant differences were established between the compared groups (Table 8).

No statistically significant difference between the body composition analysis and smoking cigarettes (Appendix A) was found.

A statistically significant relationship between alcohol consumption and the phase angle value was established. The value of this parameter was significantly lower in the group that declared drinking alcohol than in those who denied using this consuming alcohol (Mdn = 5.5 vs. 5.7; *p* = 0.005; η^2^ = 0.041). All the other parameters showed no statistically significant differences between the groups (Table 9).

A vague negative correlation between the serum level of vitamin D in the studied women and adipose tissue mass (%) (rho = −0.18; *p* = 0.011), visceral adipose tissue (rho = −0.18; *p* = 0.014), BMI values (rho = −0.22; *p* = 0.002), muscle mass (rho = −0.19; *p* = 0.008), osseous tissue mass (rho = −0.18; *p* = 0.013), and phase angle values (rho = −0.2; *p* = 0.005) was found. The water content (%) was positively correlated with the serum vitamin D level (rho = 0.18; *p* = 0.01). Data analysis showed no statistically significant correlation between the serum vitamin D level and metabolic age (*p* = 0.08) (Table 10).

## 4. Discussion

The menopausal age is an important biologic factor that not only means the loss of fertility but also stands for an increased risk of developing various diseases and middle-age problems. Some of the diseases might be prevented by introducing appropriate changes in lifestyle, hormonal replacement therapy, or supplementation containing, above all, calcium, vitamin D, and microelements. Aging is frequently named as one of the causes of the increase both in adipose tissue mass and weight in middle-aged women [24,25,26,27]. The author’s own studies showed no statistically significant relationships between age and BMI, whereas age was positively correlated with visceral adipose tissue as well as metabolic age. Age was weakly negatively correlated with the water content.

According to Ahuja et al., who performed the analysis of the relationship between different factors (above all, body composition) influencing the menopausal age among more than 2000 Indian women, showed that as a woman’s age increases, the body mass and BMI also increase [28]. Sowers et al. used bioelectric impedance and noted a linear increase in the adipose tissue mass as well as a slight linear decrease in lean body mass with the increase in the age of respondents. However, they did not identify the influence of FMP time (final menopausal period) on the adipose tissue and lean body mass in 130 surveyed women [29]. Davies et al. also described a linear increase in weight over time, and they did not establish the influence of FMP on weight [30]. Pontzer et al. showed that the total daily energy expenditure reflects daily energy needs and is a critical variable in human health and physiology. Fat-free mass-adjusted expenditure remains stable in adulthood between 20 and 60 years of age, then declines in older adults. It should be taken into account when planning nutrition and health strategies in women in the menopausal period [31].

The author’s own studies indicated a relationship between muscle mass as well as the osseous tissue mass and regular menstruation cycles. Values of both parameters were higher in the group of women who menstruated regularly in comparison with those who did not. Additionally, a correlation was established between adipose tissue mass, visceral adipose tissue, metabolic age, the water content in the body, and the presence of menstruation. Higher values of all parameters were noted in the group of non-menstruating women in comparison with menstruating ones. Similar results were achieved by Dimitruk et al., who stated that the differences in the somatic build and body composition in women depended on the menstruation condition. The highest values of body mass, hip circumference, and skin-fold thickness were noted in the perimenopausal group, while women after menopause were characterized by the highest percentage of body fat (PBF) and lowest lean body mass, soft tissue and total body water content in the organism. The highest percentage of obese women was identified in the postmenopausal group, 40% of whom suffered from visceral obesity. The onset of menopause caused changes in adipose tissue distribution, causing its transition towards the torso [32]. Greendale et al., in a study on woman’s health (SWAN), assessed the body composition with dual-energy X-ray absorptiometry. It showed that an accelerated increase in adipose tissue mass and loss of lean body mass are phenomena related to the MT (menopause transition). The rate of increase in summed adipose and lean mass does not differ between the premenopausal period and the MT, so there is no notable difference in the rate of body mass increase at the beginning of the MT [33]. The analysis of Abdulnour et al.’s study results on 48 women showed that neither the mass nor BMI are dependent on the time since the FMP, and the percentage of adipose tissue was higher in the years after the FMP than before. No change in the adipose tissue percentage was observed in the transition phase before the FMP [34].

The author’s own studies did not establish a link between alcohol consumption and most body composition parameters. The only established correlation existed between drinking alcohol and the phase angle value, which is an indicator of an organism’s nourishment. The surveyed participants who declared consuming alcohol were characterized by lower values of phase angle, which might point toward low cell metabolism and might imply an organism’s exhaustion or poor condition [35].

Thomson et al.’s study of a group of 15,920 postmenopausal women with normal weight showed that women who reported moderate alcohol consumption showed a lower risk of being overweight and obese. Perhaps body mass control measures in this population should be focused on behavior other than restricting alcohol consumption in the case of individuals with normal BMI who consume moderate amounts of alcohol [36]. Similar results were obtained by Liangpunsakul, who showed in a study on 19,000 adult women that individuals who consumed more than 30 g of alcohol a day showed a 27% lower risk of being overweight or obese than the abstinent ones [37]. In the study carried out by Mozaffarian et al., it was noted that individuals who consumed alcohol more frequently had more stable weight during the 10 years of observation than the non-drinkers. In the analysis of eating habits and weight fluctuations, drinking alcohol was inversely correlated with a body mass increase [38].

Women in the perimenopausal period are exposed to vitamin D insufficiency caused by hormonal changes. A lower level of estrogen tends to cause a lower level of serous vitamin D. Hormonal changes might cause musculoskeletal and metabolic dysfunctions as well as changes in the cardiovascular system, and they may also influence mental health as a result of vitamin D deficiency. The author’s own studies showed that with the increase in BMI, the level of vitamin D decreased significantly. It was established that almost 80% of respondents demonstrated a below-the-norm level of vitamin D in the serum.

Vázquez-Lorente et al. showed that vitamin D deficiency occurred in 80% of postmenopausal women who resided in Grenada (Spain) [39]. Similarly, Li et al. observed vitamin D deficiency in 72% of healthy postmenopausal women; no relationship between 25-(OH) D concentration and BMI was noted [40]. Studies on healthy perimenopausal women in Cathar showed that the majority of respondents struggled with serious vitamin D deficiency (the average level was 19.8 ng/mL) [41]. Similar results were obtained by researchers from Brazil, who observed vitamin D deficiency in the elderly, especially in women [42]. Corresponding results were found by authors of a multiannual project which confirmed a relationship between vitamin D concentration and BMI values. In total, over 10,000 people of different ages were tested, dividing the study group based on respondents who declared vitamin D supplementation and those who received a placebo; the need for an increased dose of vitamin D supplement for obese participants was also taken into account [43]. Based on the studies of elderly individuals above the age of 65, Huang et al. showed that females were more vulnerable to low levels of vitamin D (found in almost 60% of respondents). This condition could have been influenced by insufficient exposure to sunlight and inadequate amounts of vitamin D consumed with meals [44].

Additionally, a weak negative correlation between the serum vitamin D test results and the percentage of adipose tissue mass was established in the author’s own studies. Similarly, Arunabh et al. conducted a study on a group of 400 healthy women (age average was 47.6 years) to analyze the relationship between the serum vitamin D level and percentage of adipose tissue, taking into consideration the influence of seasons, age, and the amount of vitamin D consumed with diet. It was found that the organism’s adipose tissue content was inversely proportionate to the 25-(OH) D blood concentration in respondents. This means that the adipose tissue mass is unaffected by the serous vitamin D level. A correlation between the test results and seasons was observed. The highest concentration of serum vitamin D was noted from June to September, whereas the lowest was observed from February to May. Those studies suggest that, while assessing the need for oral vitamin D supplementation, special attention should be paid to the level of the organism’s adiposity [45]. It is worth mentioning that almost 39% of women undergoing menopausal transition are either overweight or obese. The menopausal period has been identified as a high-risk stage for weight gain in a woman’s lifecycle—a weight gain of half a kilogram per year is generally observed. Menopause-related weight gain is a consequence of low circulating estrogen levels due to the progressive loss of ovarian function. The changes in the hormonal milieu, chronological aging, decline in physical activity coupled with a Westernized dietary pattern, and recurrent emotional eating episodes associated with psychological distress also contribute to an increase in weight. A slow metabolic rate in menopausal women decreases their capacity to effectively burn calories, promoting a positive calorie balance. These menopausal symptoms can be effectively managed through lifestyle modification [46]. Physical activity helps to achieve negative calorie balance and also relieves vasomotor symptoms. Menopausal women are generally advised to aim for 150 min/week of moderate-intensity physical activity: walking, jogging, swimming, and cycling. It is also advised that resistance training exercises 2–3 times/week help to maintain bone and muscle mass. Physical activity, especially a combination of aerobics, resistance training, and balance exercises, is the most effective in controlling weight, mitigating vasomotor symptoms, and reducing psychological distress [47].

The strength of this study was that factors disturbing the assessment of vitamin D saturation were offset: there was a lack of supplementation, and tests were conducted across the whole studied group as soon as possible to rule out the risk differences in vitamin D levels caused by the seasons changing. Undoubtedly, the cross-sectional type of the study, which only allowed for screening a small percentage of a specific population of women, is a limitation. However, the mentioned difficulty does not influence the credibility of the results of this study but rather emphasizes the need to expand the research on perimenopausal women.

## 5. Conclusions

Sociodemographic factors such as age and level of education may correlate with particular parameters of body composition in perimenopausal women. With the increase in age of the surveyed women, a significant increase in visceral adipose tissue and metabolic age, as well as a decrease in body water, was observed. The level of education also differentiated women in terms of, above all, adipose tissue mass, visceral adipose tissue, metabolic age, BMI, muscle mass, osseous tissue mass, water content, and phase angle values. This indicates the need to undertake actions promoting a healthy diet and physical activity in women at this stage of life.Non-menstruating women were characterized by elevated body composition parameters, especially adipose tissue mass, visceral adipose tissue, and metabolic age. Increased muscle mass and osseous tissue mass were higher in regularly menstruating women.Women who declared too frequent alcohol consumption showed lower values of phase angle, which may indicate a weaker capacity of cells to conduct metabolic processes and reduced cell membrane permeability.A correlation between vitamin D concentration and parameters of the body composition analysis in the studied women was observed.

## Figures and Tables

**Table 1 ijerph-19-15831-t001:** Characteristics of the study group considering the sociodemographic factors, smoking tobacco, alcohol consumption, and biological factors.

Variable	Factor	N	%
Level of education	Vocational	12	6.3
	secondary	74	38.7
	higher	105	55
Place of residence	Country	17	8.9
	City up to 10 thousand residents	6	3.1
	City between 10 and 100 thousand residents	20	10.5
	City with more than 100 thousand residents	148	77.5
Marital status	Formal relationship	136	71.2
	Informal relationship	24	12.6
	single	31	16.2
Professional activity	Active	168	88
	Not active	23	12
Chronic disease	Occurs	112	58.6
	Does not occur	79	41.4
Menstruation	Occurs	72	37.7
	Does not occur	119	62.3
Regular menstruating cycles	Yes	49	68.1
	No	23	31.9
Smoking cigarettes	Yes	25	13.1
	No	166	86.9
Alcohol abuse	Yes	119	62.3
	No	72	37.7

N—number of respondents; %—percentage of the whole study group.

**Table 2 ijerph-19-15831-t002:** Characteristics of the study group concerning the anthropometric measurements.

Variable	M ± SD	Mdn	Mini–Max	CV (%)
Body height (cm)	164.8 ± 5.76	164	150–180	3.5
Body mass (kg)	72.6 ± 13.94	69.8	44.4–115.4	19.2
Waist circumference (cm)	88.7 ± 12.87	89	61–129	14.5
Hip circumference (cm)	102.5 ± 10.04	100	75–136	9.8

M ± SD—mean and standard deviation; Mdn—median; Min–Max—minimum and maximum; CV—coefficient of variables.

**Table 3 ijerph-19-15831-t003:** Marked level of vitamin D in the serum of studied women (N = 191).

Vitamin D Level	N	%
Deficiency	1	0.5
Below the norm	149	78
Norm	41	21.5

N—number of respondents; %—percentage of the whole study group.

**Table 4 ijerph-19-15831-t004:** Detailed characteristics of the body composition in the studied women (N = 191).

Body Composition Analysis	M ± SD	Mdn	Mini–Max	CV (%)
Adipose tissue mass (kg)	24.7 ± 8.71	23.7	8.5–47.8	35.3
Adipose tissue mass (%)	33.1 ± 5.9	33.6	18.9–45.4	17.8
Visceral adipose tissue	7.2 ± 2.44	7	3–15	33.8
Metabolic age (years)	49.2 ± 11.41	47	30–74	23.2
Muscle mass (kg)	45.4 ± 5.6	45	34.1–64.2	12.3
Osseous tissue mass (kg)	2.4 ± 0.3	2.4	1.8–3.4	12.2
Water content (kg)	34 ± 4.22	33.5	25.3–48.2	12.4
Water content (%)	47.4 ± 4.3	47.1	32.9–57.7	9.1
Phase angle (°)	5.7 ± 0.66	5.6	4.6–10.7	11.7

M ± SD—mean and standard deviation; Mdn—median; Min–Max—minimum and maximum; CV—coefficient of variables.

**Table 5 ijerph-19-15831-t005:** The influence of age on the body composition analysis in the studied women.

Age	rho—Spearman	*p*
Adipose tissue mass (%)	0.14	0.06
Visceral adipose tissue	0.38	0.000
Metabolic age (years)	0.36	0.000
BMI (kg/m^2^)	0.11	0.129
Muscle mass (kg)	−0.02	0.762
Osseous tissue (kg)	−0.02	0.829
Water content (%)	−0.14	0.046
Phase angle (°)	−0.09	0.218

rho-Spearman—rank correlation coefficient; *p*—test probability.

**Table 6 ijerph-19-15831-t006:** Body content analysis in relation to the level of education of studied women.

Level of Education		Vocational (N = 12)		Secondary (N = 74)		Higher (N = 104)		p	η^2^
	Body Composition Analysis	Mdn	IQR	Mdn	IQR	Mdn	IQR		
Adipose tissue mass (%)		28.7	10.8	34.9	8	32.8	7	0.026	0.028
Visceral adipose tissue		6	4	8	4	6	3	0.005	0.046
Metabolic age (years)		44	20	50.5	17	4	15	0.011	0.037
BMI (kg/m^2^)		23.7	10.4	27.6	6.6	25	5.7	0.005	0.046
Muscle mass (kg)		40.5	8.8	46.7	7.2	44.2	6	0.013	0.036
Osseous tissue mass (kg)		2.2	0.5	2.5	0.4	2.4	0.3	0.01	0.038
Water content (%)		50.4	7.6	46.3	5.5	47.7	4.8	0.015	0.034
Phase angle (°)		5.8	0.5	5.7	0.5	5.5	0.7	0.042	0.023

N—number of respondents; Mdn—median; IQR—interquartile range; *p*—test probability ANOVA rank Kruskal–Wallis); η^2^—(eta-squared) effect size.

**Table 7 ijerph-19-15831-t007:** Body composition analysis in relation to the presence of menstruation among the studied women.

Presence of Menstruation		Menstruates (N = 72)		Does not Menstruate (N = 119)		*p*
	Body Composition	Mdn	IQR	Mdn	IQR	
Adipose tissue mass (%)		32.2	8.9	34.4	7.9	0.018
Visceral adipose tissue		6	2.5	8	3	0.000
Metabolic age (years)		42	17.5	49	19	0.000
BMI (kg/m^2^)		24.8	6.3	26.2	6.9	0.1
Muscle mass (kg)		44.6	7.1	45.5	7	0.672
Osseous tissue mass (kg)		2.4	0.4	2.4	0.4	0.664
Water content (%)		48.4	6.2	46.4	5.5	0.006
Phase angle (°)		5.6	0.7	5.6	0.6	0.515

N—number of respondents; Mdn—median; IQR—interquartile range; *p*—test probability (U Mann–Whitney test).

**Table 8 ijerph-19-15831-t008:** Body composition analysis considering the regularity of the menstruation cycle.

Menstruation Cycle Regularity		Regular (N = 49)		Irregular (N = 23)		*p*
	Body Composition	Mdn	IQR	Mdn	IQR	
Adipose tissue mass (%)		32.2	8.2	32.2	9.2	0.22
Visceral adipose tissue		6	3	6	3	0.368
Metabolic age (years)		43	18	42	15	0.253
BMI (kg/m^2^)		24.8	6.5	24.8	5.2	0.29
Muscle mass (kg)		45.8	7.5	43.2	6.8	0.016
Osseous tissue mass (kg)		2.5	0.4	2.3	0.4	0.031
Water content (%)		48.4	5.8	48.1	6.4	0.22
Phase angle (°)		5.7	0.7	5.5	0.6	0.296

N—number of respondents; Mdn—median; IQR—interquartile range; *p*—test probability (U Mann–Whitney test).

**Table 9 ijerph-19-15831-t009:** Body composition analysis in relation to alcohol consumption in the studied women.

Alcohol Consumption		Yes (N = 58)		No (N = 133)		*p*
	Body Composition	Mdn	IQR	Mdn	IQR	
Adipose tissue mass (%)		34	8.1	33.5	7.8	0.885
Visceral adipose tissue		7	3	7	3	0.9
Metabolic age (years)		49	18	47	18	0.486
BMI (kg/m^2^)		25.1	6.6	26.1	5.7	0.611
Muscle mass (kg)		45.8	7.5	44.4	6.6	0.455
Osseous tissue mass (kg)		2.5	0.4	2.4	0.4	0.563
Water content (%)		46.9	5.4	47.1	5.9	0.982
Phase angle (°)		5.5	0.6	5.7	0.7	0.005

N—number of respondents; Mdn—median; IQR—interquartile range; *p*—test probability (Mann–Whitney U test).

**Table 10 ijerph-19-15831-t010:** Correlation between vitamin D level and body composition parameters in studied women.

Vitamin D	rho—Spearman	*p*
Adipose tissue mass (%)	−0.18	0.011
Visceral adipose tissue	−0.18	0.014
Metabolic age (years)	−0.13	0.08
BMI (kg/m^2^)	−0.22	0.002
Muscle mass (kg)	−0.19	0.008
Osseous tissue mass (kg)	−0.18	0.013
Water content (%)	0.18	0.01
Phase angle (°)	−0.2	0.005

rho–Spearman—rank correlation coefficient; p—test probability.

## Data Availability

Not applicable.

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
