# Peer review of "Body Composition Analysis in Perimenopausal Women Considering the Influence of Vitamin D, Menstruation, Sociodemographic Factors, and Stimulants Used"

_ijerph, 2022, doi:10.3390/ijerph192315831_

Round 1

Reviewer 1 Report

Menopausal transition is important clinical issue. Total daily energy expenditure ("total expenditure") reflects daily energy needs and is a critical variable in human health and physiology, but its trajectory over the life course is poorly studied. According to Pontzer et al., 2021, Science, total expenditure increased with fat-free mass in a power-law manner, with four distinct life stages. Fat-free mass-adjusted expenditure remains stable in adulthood (20 to 60 years), even during pregnancy; then declines in older adults. These changes shed light on human development and aging and should help shape nutrition and health strategies across the life span – the data should be cited and discussed. Additionally, weak negative correlation between  the serum vitamin D test results and percentage of adipose tissue mass was established in  author’s own studies and these results should be further explained what may be important from clinical point of view.

            The problem of obesity is rising at an alarming rate, with disproportionately higher prevalence in female than male counterparts. This trend can largely be attributed to differences in age‑specific reproductive cycles (pregnancy, lactation, and menopause) in women. Almost 39% women undergoing menopausal transition are either overweight or obese. Chronological aging is a competing risk factor for weight gain in middle‑aged women. On an average, a weight gain of half kilogram per year is generally observed in this group. Moreover, weight gain coexists with a decline in lean mass. Slow metabolic rate in menopausal women decreases their capacity to effectively burn calories promoting a positive calorie balance. Other physiological causes such as hypothyroidism, polycystic ovary syndrome, and musculoskeletal disorders can act as reasons for weight gain in menopausal women (Chopra et al., 2019. Therefore, the role of physical activity should be mentioned.

Author Response

November 8, 2022

Dear Sir or Madam,

We are very grateful for the review of our article titled “ Body composition analysis in perimenopausal women considering the influence of vitamin D, menstruation, sociodemographic factors, and stimulants used ”.

We would like to thank you for all your comments and suggestions, which helped us to improve our manuscript.

The following corrections have been introduced in order to address the suggestions of the Reviewer 1:

Menopausal transition is important clinical issue. Total daily energy expenditure ("total expenditure") reflects daily energy needs and is a critical variable in human health and physiology, but its trajectory over the life course is poorly studied. According to Pontzer et al., 2021, Science, total expenditure increased with fat-free mass in a power-law manner, with four distinct life stages. Fat-free mass-adjusted expenditure remains stable in adulthood (20 to 60 years), even during pregnancy; then declines in older adults. These changes shed light on human development and aging and should help shape nutrition and health strategies across the life span – the data should be cited and discussed.

Additionally, weak negative correlation between  the serum vitamin D test results and percentage of adipose tissue mass was established in  author’s own studies and these results should be further explained what may be important from clinical point of view.

 The problem of obesity is rising at an alarming rate, with disproportionately higher prevalence in female than male counterparts. This trend can largely be attributed to differences in age‑specific reproductive cycles (pregnancy, lactation, and menopause) in women. Almost 39% women undergoing menopausal transition are either overweight or obese. Chronological aging is a competing risk factor for weight gain in middle‑aged women. On an average, a weight gain of half kilogram per year is generally observed in this group. Moreover, weight gain coexists with a decline in lean mass. Slow metabolic rate in menopausal women decreases their capacity to effectively burn calories promoting a positive calorie balance. Other physiological causes such as hypothyroidism, polycystic ovary syndrome, and musculoskeletal disorders can act as reasons for weight gain in menopausal women (Chopra et al., 2019. Therefore, the role of physical activity should be mentioned.

Manuscript has been supplemented with the contents suggested by the Reviewer. We would like to thank you for comments and suggestions.

Kindest regards

Daria Schneider-Matyka

Reviewer 2 Report

The present cohort based study included 191 women’s where the impact of menstruation, vitamin D concentration on women’s health. The study was conducted by measuring serum vitamin D levels and adipose tissue mass. Menstruating women showed increased adipose tissue, visceral adipose tissue  and muscle mass. Reduced level of Vitamin D identified in the subjects.

Major comments-

1.     Previous studies on vitamin D levels in menstruating women in different cohorts across the globe must be included in the introduction section.

2.     Exclusion criterion for selecting subjects must be mentioned.

3.     Description of method on adipose tissue analysis must be elaborated.

4.     Statistical method used needs to be mentioned in detail.

Author Response

November 8, 2022

Dear Sir or Madam,

We are very grateful for the review of our article titled “ Body composition analysis in perimenopausal women considering the influence of vitamin D, menstruation, sociodemographic factors, and stimulants used ”.

We would like to thank you for all your comments and suggestions, which helped us to improve our manuscript.

The following corrections have been introduced in order to address the suggestions of the Reviewer 2:

The present cohort based study included 191 women’s where the impact of menstruation, vitamin D concentration on women’s health. The study was conducted by measuring serum vitamin D levels and adipose tissue mass. Menstruating women showed increased adipose tissue, visceral adipose tissue  and muscle mass. Reduced level of Vitamin D identified in the subjects.

Major comments-

  1. Previous studies on vitamin D levels in menstruating women in different cohorts across the globe must be included in the introduction section.

Manuscript has been supplemented with the contents suggested by the Reviewer.

  1. Exclusion criterion for selecting subjects must be mentioned.

We added the missing information.

  1. Description of method on adipose tissue analysis must be elaborated.

We added the missing information.

  1. Statistical method used needs to be mentioned in detail.

Description of statistical methods  has been supplemented.

We would like to thank you for comments and suggestions.

Kindest regards

Daria Schneider-Matyka

Round 2

Reviewer 2 Report

I highly appreciate the authors to answer all the concerns.